# Pharmacokinetics of Vancomycin Installation in Pleural Cavity—A Clinical Case with Animal Experiments

Soojin Lee [1], Hyo Yeong Ahn [1],*, Keunyoung Kim [2], Jeong Hun Kim [2], Soo Young Moon [3] and Yeong Dae Kim [1]

1 Department of Thoracic and Cardiovascular Surgery, Pusan National University Hospital, Busan 49241, Korea; lsj31514401@gmail.com (S.L.); domini@pusan.ac.kr (Y.D.K.)
2 Department of Nuclear Medicine, School of Medicine, Biomedical Research Institute, Pusan National University Hospital, Busan 49241, Korea; buisket@naver.com (K.K.); kmky82@naver.com (J.H.K.)
3 Department of Laboratory Medicine, Seoul Medical Center, Seoul 02053, Korea; symoon9@gmail.com
* Correspondence: doctorahn02@hanmail.net; Tel.: +82-10-4012-8202

**Abstract:** (1) background: Postpneumonectomy empyema is often observed in patients after a complete pneumonectomy. The management of these cases can be challenging when the condition of patients is complicated by a bronchopleural fistula. A multidisciplinary approach is required to manage these critically ill patients, especially when they are not suitable candidates for surgery; (2–3) Methods & Results: we report a case of successfully treated postpneumonectomy empyema caused by a bronchopleural fistula and pharmacokinetics of vancomycin installation in pleural cavity using rat experiments; (4) Conclusions The experiments provide evidence that irrigation of the pleural cavity with an antibiotic solution containing vancomycin may be an efficient treatment strategy, especially in the case of an MRSA infection in the thickened pleura.

**Keywords:** bronchial fistula; bronchoscopy; empyema; septal occluder device; therapeutic irrigation; vancomycin

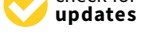



## 1. Introduction

Postpneumonectomy empyema (PPE) is a severe complication, and is often associated with a bronchopleural fistula (BPF). The operative mortality of PPE with a BPF is approximately 7.1% [1,2]. The traditional management of PPE comprises three procedures: pleural drainage, sterilisation (using an open or closed technique) and obliteration of the pleural cavity with an antibiotic solution or a muscle flap [1]. In patients who are unable to tolerate surgery, antibiotic therapy with pleural drainage through a chest tube is the recommended treatment strategy to control infection.

## 2. Case Report

In our center, a 61-year-old man underwent a complete right pneumonectomy for squamous cell carcinoma in the right lower lobe. His medical history included right upper bilobectomy for an aspergilloma four years prior. He was discharged one month following surgery with no complications. One month after discharge, he was admitted to the emergency department presenting with fever and productive sputum caused by PPE. In addition, chest computed tomography showed an air-filled space around the bronchial stump. Video-assisted thoracoscopic inspection was performed and a BPF was revealed (Figure 1A).

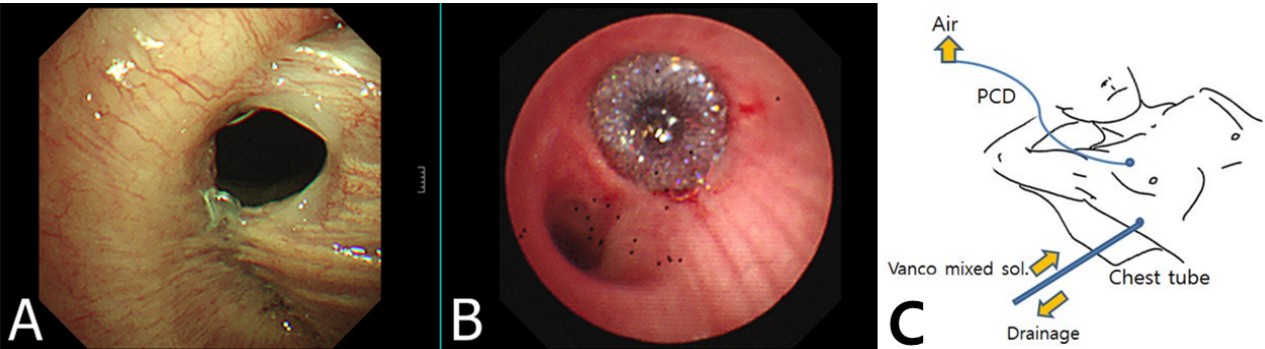

**Figure 1.** (**A**) Bronchopleural fistula before Amplatzer Septal-Occluder device placement; (**B**) Bronchopleural fistula after Amplatzer Septal-Occluder device placement; and (**C**) Antibacterial-solution irrigation as a bed-side procedure. (PCD: pigtail catheter drainage, Vanco mixed sol.: solution of 1 g vancomycin mixed with 200 mL of normal saline).

Considering the poor general condition of the patient and the state of malnutrition, we closed the BPF using an Amplatzer via fiberoptic bronchoscopy (Figure 1B). Methicillin-resistant *Staphylococcus aureus* (MRSA) was detected in the pleural fluid collected from the chest tube during the four days after Amplatzer placement. The infected pleural space was sterilised by irrigation via a chest tube using vancomycin solution (200 mL of normal saline 0.9% with 1 g of vancomycin). It took two months to eradicate the MRSA using this treatment strategy. During irrigation, an arrow catheter was placed at the third intercostal space of the midclavicular line to vent the intrathoracic air for a sufficient influx of solution (Figure 1C). The concentration of vancomycin in the pleural fluid and serum collected 6 h after an antibiotic solution irrigation was measured using therapeutic-drug monitoring. The serum concentration of vancomycin was consistently within the therapeutic range, and was recorded at 15.88 mg/dL and 18.15 ug/mL on the 40th and 45th day after Amplatzer deployment, respectively [3]. The pleural concentration of vancomycin was 27.76 μg/mL and 22.41 μg/mL on the 40th and 45th day, respectively, which was higher than the serum concentration. This vancomycin concentration was predicted to maintain over trough level. Three sessions of irrigation using video-assisted thoracic surgery and an an antibacterial solution containing vancomycin were performed.

The MRSA in the pleural effusion disappeared completely one month after the Amplatzer placement, and a bacterial culture test was performed three times, with a result of "no growth". Finally, the chest tube was removed and the patient was discharged 55 days after Amplatzer placement. In the six months following the application of the Amplatzer device, there were no other complications, and the condition of the patient improved considerably.

### 3. Materials and Methods

#### 3.1. Animals

We designed an experiment of Sprague-Dawley (SD) rats to determine the pharmacokinetics of vancomycin reabsortion into the vessels in normal and thickened pleura. SD male rats (six weeks old, weighing 200 to 250 g) purchased from KOATECH Ltd. (Gyeonggi, Korea) were employed in this study. Each group was weight matched at the start of the trial. After one week of acclimatisation, the rats were randomly divided into groups with normal pleura (n = 45) and with thickend pleura (n = 45). All rats were maintained on a 12-h light/dark cycle and provided rat chow and water *ad libitum*. Animal care and research protocols were based on the principles and guidelines adopted by the Guide for Pusan National University Hospital's Institutional Animal Care and Use Committee. All procedures were approved by the Pusan National University Hospital's Institutional Animal Care and Use Committee (No. 2020-165).

### 3.2. Pleura Thickness Rat Model Establishment

The rats were anesthetised using isoflurane inhalation (3% dissolved in oxygen), intraperitoneally received 5 mg/kg of tramadol to relieve pain. To create the thickened pleura, we intrapleurally injected Viscum (ABNOVA Viscum® Fraxini Injection, manufactured by ABNOVA GmbH, Germany) 9 mg/kg as a sclerosing agent to increase fibrinogenesis in pleural mesothelial cells. Three weeks after abnova viscum administration, the parietal pleural thickening was shown (Figure 2A,B). We decided to exclude the animal with severe infection and bleeding during procedure, and with no response after stimulation, which may have influence on the results. For each experimental group, there were no exclusions.

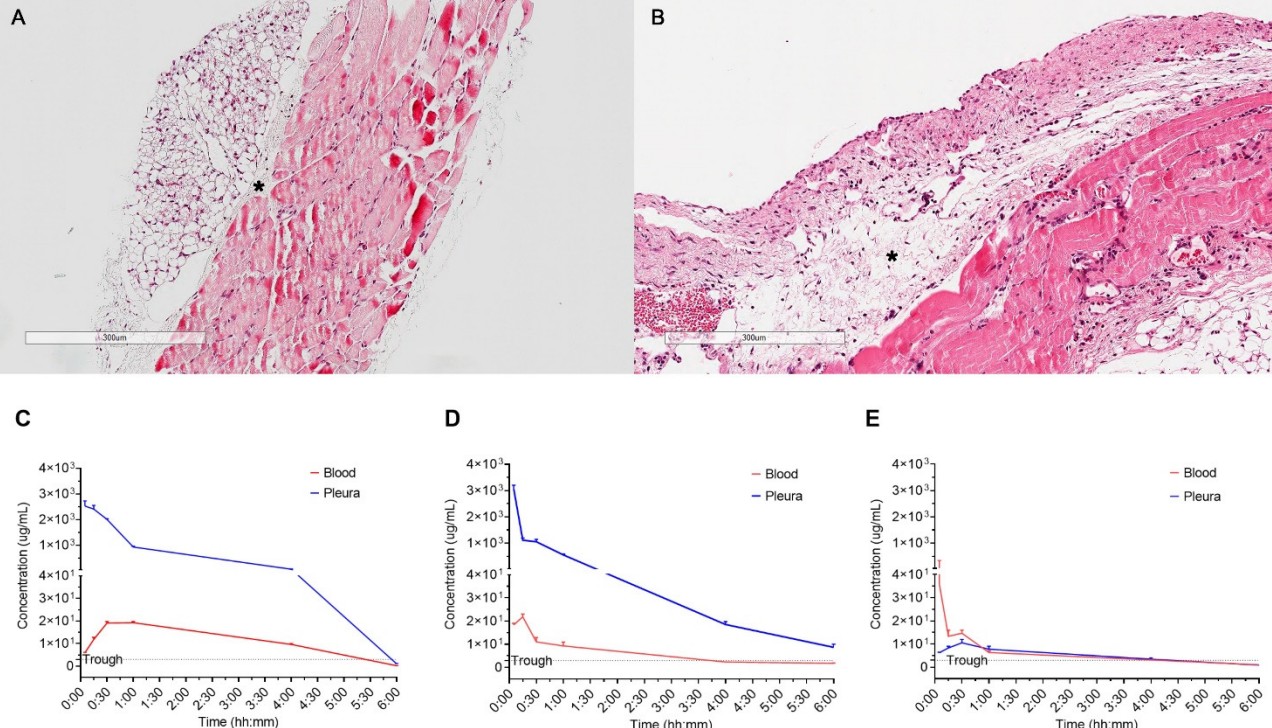

**Figure 2.** (**A**) H&E-stained histological section of the normal pleura; (**B**) H&E-stained histological section of the inflamed pleura by viscum injection; (**C**) Vancomycin concentration in pleura and blood with normal pleura after injection via pleura; (**D**) Vancomycin concentration in pleura and blood with thickened pleura after injection via pleura; and (**E**) Vancomycin concentration in pleura and blood with thickened pleura after injection via blood. * parietal pleura.

### 3.3. Experimental Procedure

Groups with normal pleura (n = 45) and with thickened pleura (n = 45) were euthanised by a schedule (5, 15, 30, 60, 90, 120, 180, 240, 360 min after vancomycin injection to pleural cavity via 24 french angiocatheter, and blood and pleural effusion were immediately achieved and measured.

In the other method, to evaluate the permeability of the injected vancomycin via vessel to penetrate the thickened pleura, as well as the permeability of pleura from the vessel, groups with normal pleura (n = 45) and with thickened pleura (n = 45) were euthanised by a same schedule after vancomycin injection to pleural cavity via 24 french angiocatheter, and blood and pleural effusion were immediately removed and measured (Figure 2E).

### 3.4. Measuring Vancomycin Concentration Levels

Blood and pleural fluid samples were obtained upon euthanasia. The obtained samples were separated at 3000 rpm for 15 min at 4 °C, and supernatants were kept at −20 °C until analysis. The vancomycin concentration was determined using chemiluminescent immunoassay (CLIA) with Architect i2000 analyser (Abbott, North Chicago, IL, USA)

according to manufacturer's processing instructions. The ARCHITECT iVancomycin assay uses a four parameter logistic curve fit (4PLC, Y-weighted) data reduction method to generate a calibration curve.

### 3.5. Histology

The parietal pleura tissue was isolated from each rat and prepared for fixation overnight in 10% neutral buffered formalin. We used automatic tissue processor for paraffin embedding (Leica, TP1020, Semi-enclosed benchtop tissue processor) and dispensing (Leica EG1150H, Heated paraffin embedding module). Cross-sections were placed on glass slides, and sections were prepared for H&E staining. For staining analyses, slides were de-paraffinised with xylene, and then hydrated through a series of washes in 100% ethanol, 85% ethanol, 75% ethanol, 50% ethanol, and finally water. We selected proper middle portion for representative figures using undertaken at $10\times$ pictures by light microscope (Leica DM4000/600 M, Versatile upright microscope for materials analysis).

### 3.6. Statistics

Categorical variables were compared using Student's *t*-test or Wilcoxon rank-sum test were used to compare continuous variables. $p < 0.05$ was considered to indicate statistical significance. Statistical analysis was performed using the R statistical program (version 3.6.2, GNU GPL) and SPSS for Windows (version 22.0, SPSS Inc., Chicago, IL, USA).

## 4. Results

### 4.1. Vancomycin Comcentration of Serum and Pleura

Vancomycin Injection to Pleural Cavity

- Compared with normal pleura, the concentration of injected vancomycin in the thickened pleura markedly decreased in 15 min, but was sustained above the trough level for four hours (Figure 2C,D)
- The peak concentration in the serum was the highest after 15 min in the thickened pleura, and 30 min in the normal pleura

## 5. Discussion

Even though the minimal inhibitory concentration (MIC) of vancomycin is sufficient in the serum, the MIC may be insufficient for reducing bacterial proliferation in the infected area. Particularly in the infected parietal pleura, the mesothelial cells that line the pleural cavity, which is an avascular area, may already be thickened, and the MIC of vancomycin may not be enough to allow sufficient vancomycin to penetrate the thickened pleura. Vancomycin is considered as "time-dependent antibiotics", which is not dependent on the "concentration"; therefore, a larger area under the curve (AUC) correlated with a superior effect [4–6].

Since there are few reports describing the pharmacokinetics of vancomycin in the pleural cavity, we measured the vancomycin serum and pleural levels in in pleural cavity after injection via pleura and blood. Our findings revealed that the concentrations of the serum and pleural fluid in 6 h after installation in pleura were at a therapeutic level. Although the mechanism of resorption of vancomycin through the pleura is unclear, the prolonged AUC of vancomycin in the pleural cavity ensured eradication of the infection without injection of additional vancomycin in the serum.

The mesothelial cells that line the pleural cavity have an important role in the filtration of the pleural fluid, resulting from the difference in the hydrostatic and colloid-osmotic pressure between the pleural fluid and capillary blood. Furthermore, it has been predicted that mesothelial cells have the ability to reabsorb antibiotics such as vancomycin, which are applied as topical agents in the thoracic cavity [7]. However, this has not been demonstrated in animal experiments.

After vancomycin injection in the pleural cavity of a rat, the phenomenon that the peak concentration in the thickened pleura reached 15 min earlier than the one in the serum

might occur because the inflammatory stimulation alters the endothelial barrier function to allow the high permeability of antibiotics in the pleura (Figure 2C,D).

Although the injected vancomycin easily penetrated the thickened pleura into the serum in the early period, the concentration of vancomycin could be maintained above the trough level in the pleural cavity with the prolonged AUC of vancomycin, which suggest that an effective topical concentration could be maintained even in thickened pleura [5]. To maintain over the trough level in pleural cavity, intrapleural installation of vancomycin needs to be repeated every four hours.

Comparing with the cases injected into the vessel, the concentration of injected vancomycin in pleura barely achieved the trough level during first one hour (Figure 2E). This phenomenon might be explained by the fact that in the thickened parietal pleura, which is an avascular area, the MIC of vancomycin may not be enough to allow sufficient vancomycin to penetrate the thickened pleura. This experiment provides evidence that irrigation of the pleural cavity with an antibiotic solution containing vancomycin may be an efficient treatment strategy, especially in the case of an MRSA infection in the thickened pleura. Owing to the difficulty in creating an animal model of chronic empyema, the efficacy of topically applied antibiotics in a thickened pleural cavity has not been sufficiently studied. In future, research should evaluate the pharmacokinetics of topically applied antibiotics in the pleural cavity of animals.

**Author Contributions:** Conceptualisation, S.L. and H.Y.A.; methodology, K.K.; formal analysis, S.Y.M.; data curation, J.H.K.; writing—original draft preparation, S.L. and H.Y.A.; supervision, H.Y.A.; and project administration, Y.D.K. H.Y.A. and K.K. contributed equally to this study. All authors have read and agreed to the published version of the manuscript.

**Funding:** This research received no external funding.

**Institutional Review Board Statement:** All procedures were approved by the Pusan National University Hospital's Institutional Animal Care and Use Committee (No. 2020-165).

**Informed Consent Statement:** Not applicable.

**Data Availability Statement:** Not applicable.

**Conflicts of Interest:** The authors declare no conflict of interest.

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
