# Peer review of "Pharmacokinetics of Vancomycin Installation in Pleural Cavity—A Clinical Case with Animal Experiments"

_applsci, doi:10.3390/app11146456_

Round 1

Reviewer 1 Report

The manuscript is not comprehensively written, as it does not describe enough methodology and does not include references. Moreover, the data were misinterpreted, and the presented data are not strong enough the authors’ conclusions.

Major comments

  1. In the case report section, the reported serum concentrations were 15.88~18.15 mg/dl, which correspond to 158.8~181.5 ug/ml. Therefore, the serum concentrations were larger, not lower than the pleural concentrations. Is it correct? Because the animal study data were opposite (pleural conc. > serum conc.).
  2. For the patent, how often was he treated with vancomycin? Describe the dosing regimen.
  3. In Figure 2, do the data represent what values? Are they mean or median? It seems like the sample size for each data point was 5. Why were not the standard deviations for each point included?
  4. There were no data on statistical analyses, which weaken the authors’ data interpretation.
  5. In discussion, the authors addressed that “after vancomycin injection in the pleural cavity of rat, the phenomenon that the peak concentration in the thickened pleura reached 15 minutes earlier than the one in the serum...”. It is not clear what it meant. Did the authors want to compare serum concentrations between normal and thickened pleura rats? Tmax values in the pleura fluid and serum should not be compared, as the drug was given into the pleural cavity.
  6. The authors claimed that “To maintain over the trough level in pleural cavity, intrapleural installation of vancomycin needs to be repeated every four hours.” However, it seems like the drug would be accumulated if vancomycin was given every 4 hours. The dosing regimen should be discussed.

Minor comment

  1. Edit the legend for Figure 2E and indicate the dosing routes for Figure 2C~E.

Reviewer 2 Report

The work described (1) a case of successful treatment of postpneumonectomy empyema by vancomycin irrigation, (2) an animal study for pharmacokinetics of vancomycin reabsorption into the vessels in normal and thickened pleura in rats. The result showed a significant difference between thickened and normal pleura in vancomycin reabsorption, which is interesting and merits publication. 

However, the manuscript is poorly written. In addition to plenty of grammar and spelling mistakes, the background and result are not scientifically described. I have several questions:

Line 51, please describe the therapeutic range and cite references.

Line 72, how many rats were excluded from each group?

Figure 2CDE, no error bar, no group number description.

Figure 2DE, what's the difference between these two figures? 

Figure 2D, the concentration in serum dropped below trough level within 2 hours, which is inconsistent with the text.

Line 141, no reference cited.

Line 152, which data lead to this conclusion?

Methods and materials: please add sample preparation and detection method for Figure 2CDE.

Reviewer 3 Report

This study described pharmacokinetics of vancomycin in postpneumonectomy empyema patients. The methods are straightforward, and the results are clearly described. Some minor issues would improve the paper. The most important result in this study is presented in the panel D and E of figure 2. By comparing those two panels, the readers can clearly understand the strength of vancomycin installization therapy in pleural cavity compared to the intravenous vancomycin injection therapy. However, it was not easy to understand the figure 2, because the panel D and E have identical explanation in the caption. The authors should add more explanation in the caption of figure 2 about the difference of panel D and E.
